

# Burnout experience among healthcare workers post third COVID-19 wave in India; findings of a cross-sectional study

Mohammad Sidiq[1], Sai Jaya Prakash Ch[2], Balamurugan Janakiraman[3,4], Aksh Chahal[1], Imran Khan[5], Surbhi Kaura[1], Faizan Kashoo[6], Farha Khan[7], Shabnam Khan[8], Chhavi Arora Sehgal[8], Shashank Baranwal[9], Sheenam Popli[10] and Mshari Alghadier[11]

[1] Department of Physiotherapy, School of Allied Health Sciences, Galgotias University, Greater Noida, Uttar Pradesh, India
[2] PDS Institute of Physiotherapy, Kaloji Narayana Rao University of Health Sciences, Purani Haveli, Hyderabad, India
[3] SRM College of Physiotherapy, SRM Institute of Science and Technology (SRMIST), Kattankulathur, Chennai, Tamil Nadu, India
[4] Faculty of Physiotherapy, School of Allied Health Sciences, Madhav University, Sirohi, Rajasthan, India
[5] Department of Physiotherapy, University of Engineering and Management, Jaipur, Rajasthan, India
[6] Department of Physical Therapy and Health Rehabilitation, College of Applied Medical Sciences, Majmaah University, Al Majmaah, Saudi Arabia
[7] Department of Dental Surgery, Northern Area Armed Forces Hospital, Hafar Al Batin, Northern, Saudi Arabia
[8] Centre for Physiotherapy and Rehabilitation Sciences, Jamia Millia Islamia University, New Delhi, India
[9] College of Physiotherapy and Occupational Therapy, Nims University, Jaipur, Rajasthan, India
[10] Department of Physiotherapy, Suresh Gyan Vihar University, Jaipur, Rajathan, India
[11] Department of Health and Rehabilitation Sciences, College of Applied Medical Sciences, Prince Sattam Bin Abdulaziz University, Alkharj, AR Riyadh Province, Saudi Arabia

Corresponding author
Mshari Alghadier,
m.alghadier@psau.edu.sa

## ABSTRACT

**Background.** The pandemic exacerbated burnout experienced by healthcare personnel, whose mental health had long been a public health concern before COVID-19. This study used the Copenhagen burnout inventory (CBI) tool to assess burnout and identify predictors among Indian healthcare workers managing COVID-19.

**Methods.** A cross-sectional study was conducted from June to December 2022, after the third pandemic wave. A web-based, fillable Google form was used to recruit COVID-19 management professionals from multiple Jaipur district hospitals. Healthcare professionals provided socio-demographic, work-related, and CBI scores. Multiple linear regression was used to control for model covariant independent variables.

**Results.** We evaluated the responses of a total of 578 participants with a mean age of $36.59 \pm 9.1$ years. Based on the CBI cut-off score of 50, 68.1% reported burnout. A total of 67.5%, 56.4%, and 48.6% of healthcare workers reported work-related, personal, and patient-related burnout, respectively. High burnout scores were significantly associated with the nursing profession ($\beta = 7.89$, 95% CI; 3.66, 12.11, $p < 0.0001$). The $p$-value indicates the probability of observing the data if the null hypothesis is true, and the confidence interval shows the range within which we can be 95% confident that the true effect lies. An independent relationship exists between male gender and higher personal-related burnout scores ($\beta = 4.45$, 95% CI 1.9–6.9).
**Conclusion**. This study identified key indicators that need further emphasis and the need for organizational and individual-level burnout monitoring in healthcare delivery sectors. Health workers continue to experience burnout due to a combination of personal, professional, and patient-related factors. This underscores the need for targeted organizational and individual interventions. The findings also suggest that the CBI tool could identify healthcare worker burnout risk groups.

## BACKGROUND

Long before the COVID-19 pandemic, the psychological well-being of healthcare professionals is a significant global public health concern (*Søvold et al., 2021*). Emotional weariness, depersonalization, cynicism, and a pessimistic assessment of one's abilities and accomplishments are all signs of burnout (*Maslach & Leiter, 2017*). Research conducted earlier than the COVID-19 pandemic showed that the prevalence of burnout in healthcare professionals varied by healthcare profession and ranged from 41% to 75% (*Bria, Baban & Dumitrascu, 2012*; *De Paiva et al., 2017*; *Dyrbye et al., 2017*; *Gosseries et al., 2012*). Physician burnout is more likely to occur when there are existential and emotional difficulties in providing care for patients who are near death (*Karlsson, Kasén & Wärnå-Furu, 2017*; *Zambrano, Chur-Hansen & Crawford, 2012*). Prior to the COVID-19 pandemic, working more than 50 hours a week, having less experience, working alone away from peers, feeling pressed for time, and lacking confidence were the main indicators of clinician burnout. However, some studies did find that critical care healthcare professionals also had an association with addressing suffering or pain, telling the family the bad news, death and dying and stopping life-sustaining treatments. Protecting the health and ability of the workforce providing healthcare necessitated recommendations and interventional strategies at the individual and organizational levels, such as exercising, building personal resilience, offering financial incentives, and proposing long-term measures to prevent clinician burnout (*Bria, Baban & Dumitrascu, 2012*; *Bienvenu, 2016*; *Luther et al., 2017*).

The COVID-19 pandemic caused significant disruptions to almost every endeavor worldwide, with the psychological well-being of healthcare professionals suffering the most. The COVID-19 pandemic has caused changes in clinicians' work lives that have not been experienced before, accompanied by an increase in stress and burnout that corresponds with these changes. Clinicians have been confronted with new sources of stress ever since the beginning of the pandemic. These new sources include fear of the virus, an inability to modulate workload, frontline worker, changes in responsibilities concerning childcare and eldercare, requirements for care that are perceived to be ethically untenable (such as shortage of human resources and materials), and a certain degree of questioning the meaning and purpose of their work. Clinicians have experienced anxiety, depression, stress, exhaustion, and burnout as a result of these stressors, as well as an unsustainable level of

turnover and departure from their practice in different countries (*Delgado-Gallegos et al.,* *2020*; *Luceño-Moreno et al., 2020*; *Magnavita, Tripepi & Prinzio, 2020*; *Rossi et al., 2020*).

The large populous Indian subcontinent witnessed chaotic healthcare workplaces, lack of control of workload, rationing of essential services and materials, and the healthcare delivery system was put to test. As of December 2023, India reported 45 million cases, and about 0.53 million deaths from the day the first case was reported on January 2020. The accelerated spread during the second and third wave of the COVID-19 pandemic caught the Indian healthcare professional off guard and the healthcare professional had to confront professional and personal challenges. According to the Indian Medical Association (IMA) reports based on the insurance scheme covered death, about 1,956 doctors succumbed to COVID-19 infection (*Debbarma, 2023*). Healthcare workers (HCW) faced many challenges in the early stages of the pandemic due to the disease's novelty, limited treatment options, fear of infection of self and loved ones, shortages of personal protective equipment (PPE), a greater workload, and challenges making emotionally and ethically difficult triaging and resource-allocation decisions. The WHO reported that there was a massive 25% rise in anxiety and depression rates among people at the end of the first year of the COVID-19 pandemic, and clinician burnout is one of the major public health challenges that has raised in proportion after the pandemic, particularly among people employed in healthcare delivery system (*Mehta et al., 2021*). The prevalence of burnout among HCWs ranged from 10.5% to 85.2%, more the pooled proportion of the prevalence by the studies conducted during the COVID-19 pandemic and non-pandemic period were 42% and 35% respectively. The findings of this review depict that the pandemic exacerbated the already present burnout-related conditions (*Nagarajan et al., 2024*).

Considering the commonality of stress and burnout among clinicians in the recent past, the COVID-19 pandemic imposed added stress on the vital workforce of the healthcare delivery system and the need to regularly estimate the burden of burnout across various regions in India. It is important to verify the change in the pattern of variables involved, the emergence of new independent variables, and the change in the strength of the association of independent variables with clinician burnout. Hence, this cross-sectional study aims to determine the levels of stress and burnout among healthcare workers following the third wave of COVID-19. Additionally, this study aims to identify the possible predictors of clinician burnout. The findings of this study will aid in identifying the key stress indicators that could be useful to set up individual and organizational level strategies to protect the healthcare workforce now and during future surges in stress.

## METHODS

### Study design

A cross-sectional study was conducted among healthcare workers from various healthcare centers in the Jaipur division, Rajasthan state, who were engaged in COVID-19 management. This study was approved by the departmental ethics committee of physiotherapy, NIMS University, affirming the ethical standards and oversight for the research NU/NCPT/JUNE/17. This study adhered to the principles bound in the

Declaration of Helsinki. All the participants provided online consent in accordance with the General Data Protection Regulation (GDPR) guideline recommended by the European Union. Data was collected using a structured online Google form. This study is reported in accordance with the STROBE (Strengthening the Reporting of OBservational studies in Epidemiology) recommendations.

## Study area

The study population is the healthcare workers who were engaged in COVID-19 management and worked at one of the healthcare centres in the Jaipur division. The Jaipur division (administrative division) is located in the state of Rajasthan and comprises seven districts: Alwar, Dausa, Jaipur, Jaipur Rural, Kherthal, Kotputli-Behror, and Dudu. The population of Rajasthan state was 83.6 million according to the 2011 census and the metro area population of Jaipur was estimated to be 4.2 million in 2021. As of December 2023, the reported COVID-19 cases in Rajasthan state was 1.33 million and 9,742 deaths from the day the first case was reported on January 2020.

## Data collection

Healthcare professionals were initially contacted *via* email, which included an invitation to participate, an explanation of the study's purpose, and a link to the online survey. Recruitment materials highlighted the importance of their participation, confidentiality assurances, and contact information for queries. A web-based survey questionnaire was sent to the email IDs of potential healthcare professionals secured from multi-healthcare centers that admitted COVID-19 patients during the pandemic in the Jaipur division of Rajasthan state, India. The online survey questionnaire consisted of a consent form and questions seeking information related to socio-demographics, work-related characteristics, self-reported obesity, and history of COVID-19 infection. Data was collected from 07th June to 25th December 2022. A reminder was sent twice at 14-day intervals to those who had not responded by verifying the subsequent delivery to improve the response rate. Among the 1,152 emails sent, 1,109 emails were opened at least once. The web-based questionnaire for this study consisted of three domains aimed at collecting socio-demographic, work-related characteristics, self-reported height and weight, and burnout measures. The Copenhagen Burnout Inventory (CBI) is a reliable, valid, geographically widely used among healthcare workers and a free open access, easy to use (*Barton et al., 2022*) outcome tool to measure self-reported burnout (*Kristensen et al., 2005*). The three aspects of burnout that the CBI focuses on are personal, work-related, and patient-related burnout. It consists of 19 questions, and the responses are graded on a five-point Likert rating system that includes the following options: "to a very high degree", "to a high degree", "somewhat", "to a low degree", and "to a very low degree". A score of 0, 25, 50, 75, or 100 is given based on the response. The average for each dimension is then calculated to estimate the level of burnout. Any score of 50 or more was considered in this study to indicate the existence of burnout to any extent. By calculating the average of the total scores for each burnout category (personal, work-related, and patient-related), we were also able to determine the average burnout score for each participant. To further characterize the distribution across

various disciplines, burnout was further divided into three categories: moderate (score of 50–74), high (score of 75–99), and severe (score of 100). The CBI tool is a free-to-use outcome measure and this study has used it in accordance with the publishing license.

## Sample size, sampling, and eligibility criteria

We calculated the sample size using the following assumptions and formula for single population proportion: 95% confidence interval, infinite population, 5% margin of error, and an expected prevalence of 50% for the widest variability, hence the estimated sample size allowing a design effect of 2.0 for the multistage-sampling technique was $n = 768$. Considering the poor response rate for the online survey methods and based on literature recommendations for the online survey method, we assumed a 50% non-response and contingency plan and added 50% to the required power-calculated sample size. The final computed sample size 2.0 was 1,152. The list of email IDs of potential healthcare professionals was obtained from the administrative records of healthcare centers in Jaipur division, Rajasthan State. We ensured that the participants were representative of the target population by stratifying the sample based on profession, gender, and years of experience. Non-response rates were handled by sending reminder emails at two-week intervals, and data analysis was adjusted for potential response bias. The participants of this study were those who self-reported to be relatively healthy and worked as healthcare providers and manual handling of COVID-19 patients during the third wave of the pandemic and before.

## Data analysis

Statistical analyses were conducted using the Statistical Package for Social Sciences version 25, (IBM SPSS Inc., Chicago IL, USA) for Windows. The data related to socio-demographics, clinical characteristics, and CBI scores of the healthcare workers were descriptively analyzed, and categorical variables were reported as frequencies with percentages. The normality of the continuous variables (burnout scores and age) was tested using the Kolmogorov–Smirnov, and Shapiro–Wilk tests. Data were presented as mean, standard deviation, median, and IQRs. The association between the CBI scores and independent variables like; socio-demographic variables, profession, work experiences, working hours, history of COVID-19 infection, shift of work, and whether worked has frontline during COVID were tested initially using Univariate linear regression analysis. If the variables were related to the dependent variables at $\leq 0.20$, they were included in the multiple linear regression model. A final model, multiple linear regression analysis was conducted to control for co-varying independent variables in the model. An alpha ($\alpha$) of 0.05 was set as the cut-off for the level of significance. Assumptions such as linearity, homoscedasticity, and multi-collinearity were checked to ensure the appropriateness of the regression analysis. The standardized coefficient ($\beta$) with 95% CI was reported with $p$ values and coefficients of determination ($R^2$). The multi-collinearity of the independent variables for tolerance in the final model was tested using the variance inflation factor (VIF) with <5 as the cut-off point. The data set had no missing data since the online CBI survey questionnaire permitted submission only if the form was completely attended. If outliers were identified, sensitivity analyses were conducted and data analyses were

stratified across different subsamples like profession, gender, and years of experience. The gender differences and/or similarities were analyzed according to the recommendation of *Heidari et al. (2016)*.

## RESULTS

Among the 1,152 healthcare workers who received the online survey questionnaire form, 578 responded, with a response rate of 50.17%. The participants' average age was 36.59 years, and the majority of them (31.1%) were in the 30- to 39-year-old age range. Males represented almost 3/5 of the participants. The majority of those surveyed were nurses (22.7%), followed by medical residents (16.1%), doctors (13.8%), physiotherapists (12.8%), technicians (11.2%), dentistry residents (10.4%), dentists (7.8%), and surgeons (5.2%). The participants' mean experience was 7.82 years overall, with 45.5% reporting between 6 and 10 years of experience. A little over 48% of them reported working less than 48 h a week on average, and 25.6% reported that they worked more than 60 h per week. More than half of the (51.2%) respondents were single. In total, 68.1% of healthcare-related workers reported burnout, based on the CBI cut-off score of 50. Most of them (67.5%) reported 50 and higher scores in the CBI's work-related burnout domain, followed by personal (56.4%) and patient-related burnout (48.6%) burnout. Table 1.

### Regression analysis

The univariate analysis of burnout scores and independent variables of 578 healthcare workers showed significant associations with socio-demographic factors Table 2. A significant proportion of participants reported burnout: 56.4% ($n = 326$) reported personal factors related to burnout, 67.5% ($n = 390$) reported burnout related to their jobs, and 48.6% ($n = 281$) reported burnout related to their patients, and 68.2% ($n = 394$) reported average exhaustion. Personal burnout was experienced by 48.3% of women and 61.8% of men, with a significant gender disparity ($p = 0.002$). Age was a significant factor, with the highest percentages of burnout being related to work (76.7%) and personal (81.7%) among individuals aged 30-39 ($p = 0.062$ and $p = 0.043$, respectively). Burnout was highly influenced by profession in all domains ($p = 0.001$ for personal burnout, $p = 0.000$ for work-related and patient-related burnout), with higher percentages of personal burnout seen in nurses (72.5%), physiotherapists (75.7%), and medical residents (66.7%). Patient-related burnout was highly impacted by marital status ($p < 0.001$), with divorced or widowed people experiencing a greater incidence (60.7%). Figure 1 shows the distribution of burnout categories according to the professions.

The univariate linear regression of burnout scores and work-related factors showed that the healthcare workers who worked for longer hours had work-related, personal burnout ($p$ 0.000 and $p$ 0.038 respectively) and overall burnout ($p$ 0.036). There was no correlation between any domain of burnout and working shift or front-line work. Those who had more work experience ($p < 0.001$ for personal burnout, $p = 0.032$ for work-related burnout, $p = 0.007$ for patient-related burnout) and healthcare workers with a history of COVID-19 infection ($p$ 0.03 personal burnout, $p$ 0.048 work-related burnout, $p$ 0.015

**Table 1  Socio-demographic characteristics of study participants ($n = 578$).**

| Variable | Frequency (n) | % |
|---|---|---|
| Age, mean (SD) | 36.59 (9.1) | |
| Age (years) | | |
|     <30 | 154 | 26.6 |
|     30–39 | 180 | 31.1 |
|     40–49 | 163 | 28.2 |
|     >50 | 81 | 14 |
| Sex | | |
|     Male | 348 | 60.2 |
|     Female | 230 | 39.8 |
| BMI category | | |
|     Normal weight | 408 | 70.5 |
|     Overweight | 140 | 24.3 |
|     Obese | 30 | 5.2 |
| Number of children | | |
|     None | 447 | 77.3 |
|     1 | 112 | 19.4 |
|     2 & more | 19 | 3.3 |
| Medical profession | | |
|     Surgeon | 30 | 5.2 |
|     Physician | 80 | 13.8 |
|     Medical resident | 93 | 16.1 |
|     Dentist | 45 | 7.8 |
|     Dentistry resident | 60 | 10.4 |
|     Nurse | 131 | 22.7 |
|     Physiotherapist | 74 | 12.8 |
|     Technician | 65 | 11.2 |
| Work experience, mean (SD) | 7.82 (4.19) | |
| Work experience (years) | | |
|     1–5 | 198 | 34.3 |
|     6–10 | 263 | 45.5 |
|     11–15 | 85 | 14.7 |
|     16–20 | 32 | 5.5 |
| Working hours/week | | |
|     $\leq 48$ | 278 | 48.1 |
|     49–60 | 152 | 26.3 |
|     >60 | 148 | 25.6 |
| Marital status | | |
|     Unmarried | 296 | 51.2 |
|     Married | 254 | 43.9 |
|     Divorced | 28 | 4.8 |
| Burnout (CBI score 50 & above) | | |
|     Personal | 326 | 56.4 |

**Table 1** (*continued*)

| Variable | Frequency (n) | % |
|---|---|---|
| Work related | 390 | 67.5 |
| Patient related | 281 | 48.6 |
| Average burnout | 394 | 68.2 |
| CBI Burnout | Median (Q1, Q3) | |
| Average | 65.27 (43.7, 79.2) | |
| Personal | 62.5 (38.5, 85.5) | |
| Work related | 61.5 (41.66, 87.5) | |
| Patient related | 58.33 (37.5, 83.3) | |

**Notes.**

CBI, Copenhagen Burnout Inventory; SD, Standard deviation; Q1, first quartile; Q3, third quartile.

for patient-related burnout) had significantly higher levels of burnout in all the domains (Table 3).

All the independent variables, such as sex, marital status, occupation, age group, history of COVID-19 infection, work experience, and working hours, were included in a comprehensive multiple linear regression model (Table 4). There was an independent relationship between a higher average personal burnout score and being male ($\beta = 4.45$, 95% CI; 1.9–6.9). The average burnout scores for surgeons and physicians were significantly lower for personal ($\beta = -39.2$, 95% CI; -48.15, $-30.3$ and $\beta = -14.3$, 95% CI; -20.1, $-8.57$), work-related ($\beta = -37.1$, 95%CI; $-46.1$, -28.2 and $\beta -13.1$, 95%CI; $-18.9$, $-7.26$), patient-related ($\beta -39.7$, 95% CI; $-49.23$, $-30.3$ and $\beta -16.9$, 95% CI; $-23.1$, $-10.88$), and overall ($\beta = -35.13$, 95%CI; $-43.1$, $-27.17$ and $\beta = -12.8$, 95%CI; $-17.9$, $-7.67$). Nurses, on the other hand, experienced higher average burnout and burnout related to their jobs ($\beta = 10.4$, 95% CI; 5.22, 14.86, $p < 0.0001$ and $\beta = 7.89$, 95% CI; 3.66, 12.11, $p < 0.0001$, respectively). Healthcare professionals who had previously contracted COVID-19 had greater average burnout scores ($\beta = 8.87$, 95% CI; 3.99, 13.75, $p < 0.001$), work-related burnout scores ($\beta = 7.51$, 95% CI; 1.94, 13.07, p 0.029), and personal burnout scores ($\beta = 6.13$, 95%CI; 1.06, 11.6, $p0.029$). Table 4 presents additional variables connected to the shifts in the burnout measures. In the final model, the independent variable explained the proportion of variance for personal, work-related, patient-related, and average burnout, which were $R^2$ 0.62 (62%), $R^2$ 0.69 (69%), $R^2$ 0.71 (71%), and $R^2$ 0.70 (70%) respectively.

## DISCUSSION

The purpose of the study was to assess the prevalence of burnout among Indian healthcare professionals in the wake of the COVID-19 pandemic's third wave. The research findings demonstrated a significant degree of burnout in the overall CBI average score and all three sub-domains of CBI among healthcare professionals, highlighting the stress this vital workforce endured during the crisis. The results of this study endorse that burnout was significantly more common among healthcare professionals across the domains, with rates for personal burnout being 56.4%, work-related burnout being 67.5%, and patient-related burnout being 48.6%. These findings highlight the high levels of burnout that

**Table 2  Univariate analysis of burn-out and socio-demographic characteristics.**

| Variables | n = 578 | Number of respondents with burn-out n(%) | | | |
|---|---|---|---|---|---|
| | | Personal burn-out | Work-related burn-out | Patient related burn-out | Average burn-out |
| Total (n = 578) | | 326 (56.4) | 390 (67.5) | 281 (48.6) | 394 (68.2) |
| Sex | | | | | |
| Male | 348 (60.2) | 215 (61.8) | 231 (66.4) | 164 (47.1) | 234 (67.2) |
| Female | 230 (39.8) | 111 (48.3) | 159 (69.1) | 117164 (50.9) | 160 (69.6) |
| p value | | 0.002 | 0.092 | 0.213 | 0.55 |
| Age | | | | | |
| <30 | 154 (26.6) | 45 (29.2) | 89 (57.8) | 58 (37.7) | 77 (50) |
| 30–39 | 180 (31.1) | 147 (81.7) | 138 (76.7) | 100 (55.6) | 148 (82.2) |
| 40–49 | 163 (28.2) | 88 (54) | 112 (68.7) | 83 (50.9) | 115 (70.6) |
| >50 | 81 (14) | 46 (56.8) | 51 (63) | 40 (49.4) | 54 (66.7) |
| p value | | 0.062 | 0.043 | 0.011 | 0.000 |
| Profession | | | | | |
| Surgeon | 30 (5.2) | 5 (16.7) | 8 (26.7) | 5 (16.7) | 7 (23.3) |
| Physician | 80 (13.8) | 34 (42.5) | 45 (56.2) | 25 (31.2) | 42 (52.5) |
| Medical resident | 93 (16.1) | 60 (64.5) | 62 (66.7) | 53 (57) | 64 (68.8) |
| Dentists | 45 (7.8) | 19 (42.2) | 30 (66.7) | 23 (51.1) | 28 (62.2) |
| Dentistry resident | 60 (10.4) | 40 (66.7) | 46 (76.7) | 31 (51.7) | 44 (73.3) |
| Nurses | 131 (22.7) | 95 (72.5) | 110 (84) | 76 (58) | 115 (87.8) |
| Physiotherapist | 74 (12.8) | 27 (36.5) | 56 (75.7) | 43 (58.1) | 50 (67.6) |
| Technicians | 65 (11.2) | 46 (70.8) | 33 (50.8) | 25 (38.5) | 44 (67.7) |
| p value | | 0.001 | 0.000 | 0.000 | 0.000 |
| Number of children | | | | | |
| None | 447 (77.3) | 258 (77.3) | 304 (68) | 216 (48.3) | 308 (68.9) |
| 1 | 112 (19.4) | 56 (50) | 72 (64.3) | 56 (50) | 72 (64.3) |
| 2 & more | 19 (3.3) | 12 (63.2) | 14 (73.7) | 9 (47.4) | 14 (73.7) |
| p value | | 0.382 | 0.48 | 0.94 | 0.561 |
| Marital status | | | | | |
| Not married | 228 (39.4) | 100 (43.9) | 133 (58.3) | 84 (36.8) | 132 (57.9) |
| Married | 322 (55.7) | 208 (64.6) | 240 (74.5) | 184 (57.1) | 245 (76.1) |
| Divorced/widowed | 28 (4.8) | 18 (64.3) | 17 (60.7) | 13 (46.4) | 17 (60.7) |
| p value | | 0.295 | 0.673 | 0.000 | 0.000 |
| BMI category | | | | | |
| Normal weight | 408 (70.5) | 216 (52.9) | 269 (65.9) | 190 (46.6) | 270 (66.2) |
| Overweight | 140 (24.3) | 91(65) | 101 (72.1) | 78 (55.7) | 103 (73.6) |
| Obese | 30 (5.2) | 19 (63.3) | 20 (66.7) | 13 (43.3) | 21 (70) |
| p value | | 0.37 | 0.33 | 0.44 | 0.262 |

**Notes.**
BMI,  Body Mass Index.

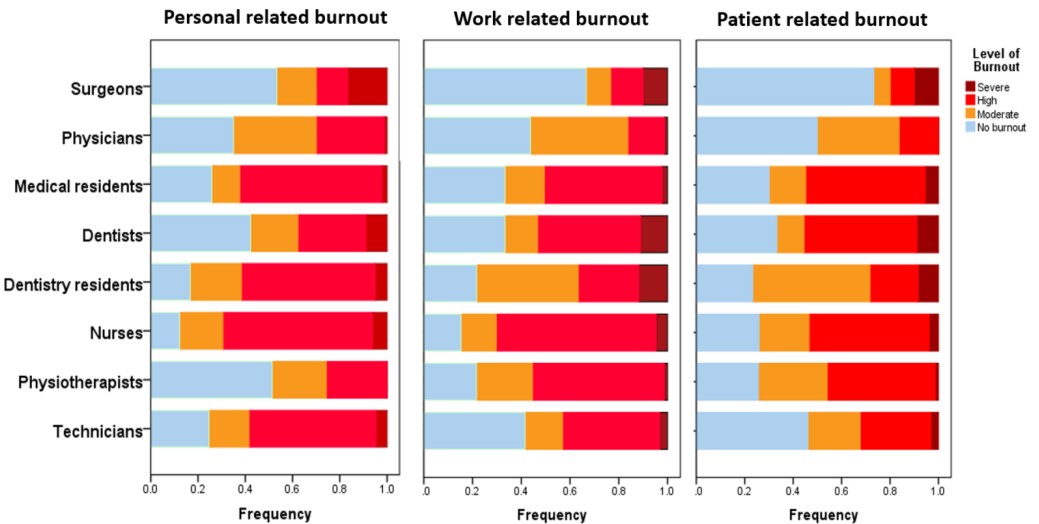

**Figure 1 Distribution of burnout according to professions.**

are encountered in the healthcare industry, which is particularly concerning considering the significant burden that the COVID-19 pandemic has placed on India's healthcare workforce and infrastructure.

This study observed a high level of burnout among healthcare workers, with significant proportions reporting burnout in personal, work-related, and patient-related domains. This highlights the toll that the pandemic has taken on the mental and emotional well-being of healthcare professionals. The prevalence of burnout syndrome among healthcare workers, as indicated by the findings of this study, is notably high, with 68.1% of respondents reporting symptoms consistent with burnout. This finding underscores the significant toll that the COVID-19 pandemic has taken on the mental and emotional well-being of frontline healthcare providers in India. The prevalence of burnout was particularly pronounced in the work-related domain, affecting 67.5% of participants, indicating the strain and challenges associated with their professional roles and responsibilities. Personal burnout was also prevalent, affecting 56.4% of respondents, reflecting the impact of work-related stressors on individual healthcare workers' mental health and personal lives. Moreover, nearly half (48.6%) of the surveyed healthcare workers reported experiencing burnout related to patient care, highlighting the emotional burden and exhaustion associated with providing care during the pandemic. Conversely, a study conducted among healthcare workers in Ghana reported a prevalence of 20.57%, notably higher than previous reports (*Konlan et al., 2022*). Non-clinicians exhibited higher burnout rates compared to clinicians (26.74% *vs.* 15.64%, $p < 0.001$). Those with 1–5 years of experience were 26.81 times more likely to experience burnout (AOR = 26.81, CI = 6.37–112.9) (*Søvold et al., 2021*). Night shifts (8:00 pm to 8:00 am) were associated with 1.86 times higher odds of burnout (OR = 1.86; 95% CI [1.33–2.61]; $p < 0.001$). Participants from primary-level facilities were 3.91 times more likely (AOR = 3.91, 95% CI = 2.39–6.41) to experience burnout. Additionally, those intending to leave their current jobs were 4.61 times more likely (AOR = 4.61, 95% CI

**Table 3 Univariate analysis of burn-out, work-related characteristics, and COVID-19.**

| Variables | n = 578 | No of respondents with burn-out | | | |
|---|---|---|---|---|---|
| | | Personal burn-out(%) | Work-related burn-out (%) | Patient related burn-out (%) | Average burn-out |
| Total (n = 578) | | 326 (56.4) | 390 (67.5) | 281 (48.6) | 394 (68.2) |
| Working hours/week | | | | | |
| ≤48 h | 278 (48.1) | 148 (53.2) | 175 (62.9) | 127 (45.7) | 176 (63.3) |
| 49–60 h | 152 (26.3) | 100 (65.8) | 110 (72.4) | 79 (52) | 107 (70.4) |
| >60 h | 148 (25.6) | 106 (71.6) | 105 (70.9) | 75 (50.7) | 111 (75) |
| p value | | **0.000** | **0.038** | 0.261 | **0.036** |
| Work shifts | | | | | |
| Day shift | 509 (88.1) | 308 (60.5) | 341 (67) | 245 (48.1) | 347 (68.2) |
| Alternate shift | 69 (11.9) | 46 (66.7) | 49 (71) | 36 (52.2) | 47 (68.1) |
| p value | | 0.358 | 0.517 | 0.308 | 0.546 |
| Work experience (years) | | | | | |
| 1–5 | 199(34.4) | 104 (61.1) | 120 (60.3) | 79 (39.7) | 116 (58.3) |
| 6–10 | 262(45.3) | 183 (63.1) | 197 (75.2) | 144 (55) | 203 (77.5) |
| 11–15 | 85(14.7) | 48 (56.5) | 55 (64.7) | 45 (52.9) | 55 (64.7) |
| ≥ 16 | 32(5.5) | 19 (59.4) | 18 (56.2) | 13 (40.6) | 20 (62.5) |
| p value | | **0.001** | **0.032** | **0.007** | **0.000** |
| Worked as frontline | | | | | |
| Yes | 219 (37.9) | 140 (63.9) | 147 (67.1) | 108 (49.3) | 154 (70.3) |
| No | 359 (62.1) | 214 (59.6) | 243 (67.7) | 173 (48.2) | 240 (66.9) |
| p value | | 0.17 | 0.47 | 0.43 | 0.219 |
| COVID-19 positive (previous/current) | | | | | |
| Yes (tested) | 83 (14.4) | 59 (71.1) | 63 (75.9) | 50 (60.2) | 66 (79.5) |
| No | 495 (85.6) | 295 (59.6) | 327 (66.1) | 231 (46.7) | 328 (66.3) |
| p value | | **0.03** | **0.048** | **0.015** | **0.010** |

**Notes.**
The bold values are significant p values (<0.05), $R^2$ value for (70.5) the model explains 70.5% variation in personal CBI score.

= 2.73–7.78) to experience burnout, while those perceiving high workloads were 2.38 times more likely (AOR = 2.38, 95% CI = 1.40–4.05) to experience burnout (*Konlan et al., 2022*). Similar, results from Romanian medical students (*Dimitriu et al., 2020*), healthcare professionals in Spain (*Torrente et al., 2021*), medical staff (*Huo et al., 2021*), physicians in teaching hospitals (*Appiani et al., 2021*), and intensive care unit specialists (*Azoulay et al., 2020*).

Burnout was found to be associated with some of the demographic factors. Significant gender disparities were observed, as males reported higher levels of personal burnout in comparison to female healthcare workers. The age of healthcare professionals appears to be a notable determinant, as individuals belonging to the 30–39 years age group exhibited the most pronounced levels of burnout. In addition, the marital status and occupation of healthcare professionals were found to have a substantial bearing on the degree of burnout they encountered. A significant portion of the participants indicated that they had encountered burnout in all 3 categories; 56.4% reported burnout associated with personal

**Table 4  Multiple linear regression for the three burn-out domains, average CBI score, and independent variables.**

| Dependent variables | Independent variables | B | 95% CI | *p*value |
|---|---|---|---|---|
| Personal burn-out R² 0.62 | Male, gender | 4.45 | 1.9, 6.9 | 0.034 |
| | Profession | | | |
| | Surgeon | −39.2 | −48.15, −30.3 | 0.000 |
| | Physician | −14.3 | −20.1, −8.57 | 0.000 |
| | Dentist | −14.8 | −22.1, −7.5 | 0.000 |
| | Physiotherapist | −17.7 | −23.6, −11.8 | 0.000 |
| | Work experience | | | |
| | 6 to 10 years | 7.9 | 3.9, 11.89 | 0.010 |
| | Age category | | | |
| | 30 to 39 years | −5.54 | −9.75, −1.13 | 0.013 |
| | COVID-19 positive | 6.13 | 1.06, 11.6 | 0.029 |
| Work-related burnout R² 0.69 | Profession | | | |
| | Surgeon | −37.1 | −46.1, −28.2 | 0.000 |
| | Physician | −13.1 | −18.9, −7.26 | 0.000 |
| | Nurse | 10.04 | 5.22, 14.86 | 0.000 |
| | Work experience | | | |
| | 6 to 10 years | 8.21 | 4.29, 12.13 | 0.000 |
| | COVID-19 positive | 7.51 | 1.94, 13.07 | 0.008 |
| Patient related burnout R² 0.71 | Age category | | | |
| | <30 years | 9.56 | 3.15, 15.97 | 0.003 |
| | Marital status | | | |
| | Married | 6.21 | 1.75, 10.7 | 0.006 |
| | Profession | | | |
| | Surgeon | −39.75 | −49.23, −30.3 | 0.000 |
| | Physician | −16.96 | −23.1, −10.88 | 0.000 |
| | Technician | −8.24 | −14.8, −1.62 | 0.015 |
| | Work experience | | | |
| | 1 to 5 years | −9.56 | −15.15, −3.98 | 0.001 |
| Average burn-out R² 0.70 | Age category | | | |
| | 30 to 39 years | −6.18 | −9.94, −2.43 | 0.001 |
| | Profession | | | |
| | Surgeon | −35.13 | −43.1, −27.17 | 0.000 |
| | Physician | −12.8 | −17.9, −7.67 | 0.000 |
| | Nurse | 7.89 | 3.66, 12.11 | 0.000 |
| | Work experience | | | |
| | 6 to 10 years | 9.16 | 5.61, 12.71 | 0.000 |
| | COVID-19 positive | 8.87 | 3.99, 13.75 | 0.000 |

**Notes.**
$\beta$ Standardized coefficients beta, R square value of $R^2$ 0.62, $R^2$ 0.69, $R^2$ 0.71, and $R^2$ 0.70 explained 62%, 69%, 71% and 70% variations in personal related burnout, work related burnout, patient related burnout, and average burnout by the independent variables respectively.

factors, 67.5% reported burnout related to job-related factors, and 48.6% reported burnout associated with patient care. Additionally, 68.2% of the participants reported experiencing overall exhaustion.

We observed gender disparities in this study, as men exhibited higher levels of personal burnout in comparison to women, thereby highlighting a substantial gender divide. Factors such as financial challenges, stress related to parenting, relationships, job roles, societal expectations, and coping mechanisms might have contributed to the higher reporting of physical burnout levels among male healthcare workers (*Zhang et al., 2022*). These findings challenge the common reporting that female employees are more likely to experience burnout than male employee (*Artz, Kaya & Kaya, 2022*). Our study's finding that the lower age group who by implication might have lesser experience in healthcare is associated with higher odds of reporting patient-related burnout, which is in agreement with several previous studies (*Delgado-Gallegos et al., 2020*; *Jalili et al., 2021*; *Galanis et al., 2021*). It is also appropriate that younger healthcare professionals would have experienced more stress due to workload, lack of coping, and lower level of exposure to challenging situations and lower clinical experience. Further, our study revealed that the middle-age group reported experiencing higher personal burnout and overall burnout. The possible explanations might be that the middle-aged respondents are more likely to encounter life-stage-related stress, family safety concerns, and job stress (*Ahola et al., 2008*). The impact of the profession on burnout is significant across various domains, with nurses, physiotherapists, and medical residents demonstrating elevated levels of personal burnout. The incidence of patient-related burnout was found to be significantly influenced by marital status, with divorced or widowed individuals exhibiting a higher prevalence. The results of this study emphasize the intricate relationship between socio-demographic factors and burnout among healthcare workers, emphasizing the necessity domains of implementing focused interventions that are customized to address the unique needs of different demographic and professional cohorts. Figure 1 provides a visual representation of how burnout categories are distributed among different professions, highlighting the differences in the prevalence of burnout within the healthcare workforce. A study conducted to examine the relationship between burnout syndrome among physicians and various individual factors found that older age, possession of an academic title, longer tenure in the profession, and longer duration at the institution were associated with higher levels of burnout. Furthermore, it underscored the significance of taking into account the enduring ramifications of choosing a specific specialty in relation to burnout (*Ozkula & Durukan, 2017*). A research investigation carried out among nurses in Iran unveiled that there were moderate levels of emotional exhaustion and depersonalization, accompanied by low levels of personal accomplishment (*Rashedi, Rezaei & Gharib, 2014*).

In a research study involving a sample of 1,830 nurses in Singapore, it was found that 39% of the participants displayed elevated levels of emotional exhaustion (EE), 40% exhibited heightened levels of depersonalization (DP), and 59% reported experiencing low personal accomplishment (PA) based on the cutoff scores of the Maslach Burnout Inventory (MBI) (*Tan et al., 2020*). The findings of the Iranian and Singapore studies are similar to the findings of our study but the outcome tools used were different. There

exists a significant correlation between burnout and work-related factors, specifically working hours and work experience. Extended periods of work were associated with elevated levels of burnout in all areas, underscoring the significance of effectively managing workload and guaranteeing sufficient rest for healthcare professionals. Furthermore, it was observed that individuals with greater professional experience exhibited elevated levels of burnout, indicating that extended exposure to demanding work settings could intensify burnout (*Konlan et al., 2022*; *Torrente et al., 2021*; *Agrawal et al., 2024*; *Navinés et al., 2021*). The data consistently indicates that surgeons and physicians exhibit lower levels of burnout in all areas when compared to other professions, as evidenced by significant negative regression coefficients. This implies that individuals occupying these positions may possess more effective coping strategies or work settings that alleviate burnout. In contrast, nurses consistently demonstrated elevated levels of burnout, specifically in relation to work-related burnout, suggesting that the requirements and pressures linked to nursing positions may contribute to increased rates of burnout. The levels of burnout among physiotherapists and technicians were found to be higher, although to a lesser degree in comparison to nurses.

The aforementioned findings highlight the significance of acknowledging the distinct obstacles encountered by various healthcare professions and implementing focused interventions to effectively tackle burnout within each occupational cohort. In addition, interventions should prioritize enhancing work conditions, offering sufficient support, and cultivating coping mechanisms customized to the unique requirements of each occupation in order to mitigate burnout and promote overall well-being among healthcare professionals. A study conducted with a sample size of 237 nurses revealed a positive correlation between work demands and both burnout syndrome and musculoskeletal complaints (*Jaworek et al., 2010*). Increased work stimuli were linked to decreased burnout, but were also associated with increased musculoskeletal complaints. A research investigation carried out among nurses revealed that there exists a correlation between positive work-related relationships and specific psychological factors, such as communication skills and empathy, which serve as protective factors against burnout (*Pérez-Fuentes et al., 2019*). Healthcare professionals with prior COVID-19 infection have reported elevated levels of burnout, underscoring the heightened burden experienced by individuals directly affected by the virus. This highlights the necessity of implementing comprehensive support systems and allocating resources to effectively tackle the mental health difficulties encountered by frontline workers both during and in the aftermath of the pandemic. The $R^2$ values denote the coefficient of determination, which signifies the extent to which the independent variables incorporated in each model can account for the variability observed in burnout scores. In the instance of personal burnout, the coefficient of determination ($R^2$) of 0.62 indicates that around 62% of the variance in personal burnout scores among healthcare professionals can be explained by variables such as gender, occupation, professional background, age group, and COVID-19 positivity. In the aforementioned models, the $R^2$ values of 0.69, 0.71, and 0.70 for work-related burnout, patient-related burnout, and average burnout, respectively, indicate the extent to which the independent variables account for

the variance observed. The R2 values offer valuable insights into the combined predictive capability of the models in comprehending burnout among healthcare professionals.

A comprehensive analysis revealed several crucial factors linked to a significant occurrence of burnout, such as being female, having less experience, not having children, and being single. These factors are particularly associated with heightened levels of anxiety, depression, and stress among women. Major findings indicated that physicians in the Republic of Korea experienced a burnout rate of 90.4% (*Park et al., 2020*), psychiatrists in Saudi Arabia had a burnout rate of 80.2% (*Alkhamees et al., 2021*), senior and specialist physicians in Ireland had a rate of 77% (*Crudden, Margiotta & Doherty, 2023*), and emergency physicians in the USA had a rate of 74.7% (*Alanazy & Alruwaili, 2023*). Notably, similar to this study, self-reported height and weight measures are commonly used to quantify obesity and overweight in online survey designs, but they are also frequently regarded as research design limitations (*Scholes et al., 2023*). It has been associated with bias and low agreement, which could be caused by recall bias or social desirability.

Considering the non-response rate in this study and respondents who were less likely to be surveyed, the non-response bias and sampling bias, the cross-sectional design not allowing causal inferences, and not considering outcomes related to perceived stress are major limitations of this study and readers shall execute caution while interpreting the findings. However, the physical measurements were not the primary outcome measure of this study, and other research has shown that certain populations self-report with some accuracy. Furthermore, we presume that the medical professionals being the study population could have demonstrated a high digital literacy and reasonable level of agreement with the real case scenario. Though the findings of this study provide valuable insight into the predictors of burnout among HCWs in the Jaipur division, there are constraints in the generalizability of the findings to all healthcare professionals in India. Future studies aiming to include a more diverse representative sample across several regions are needed to improve the generalizability. It is important to develop strategies to implement flexible work schedules to improve work-life balance, enhance job security through stable contracts, and increase mental health support services for healthcare workers. Additionally, policy recommendations like regular mental health screenings and burnout prevention programs tailored to healthcare workers are needed.

## CONCLUSION

The higher prevalence of burnout reported in this study warrants more insight and policy adjustments regarding the working hours, workload, resource allocations, mental health, safety, and security of healthcare professionals in India. Further, the epidemic of clinician burnout even prior to the COVID-19 pandemic might have flared up the burden of burnout owing to the toll on the healthcare workers during the pandemic chaos. Thus, burnout remains highly prevalent, and personal, professional, and patient-related factors affect health workers, requiring organizational and individual interventions. More importantly, the nation's health policy must take the lessons learned from the epidemic into account in order to fortify the healthcare sector.

**List of abbreviations**

| | |
|---|---|
| **AOR** | Adjusted Odds Ratio |
| **CBI** | Copenhagen Burnout Inventory |
| **CI** | Confidence Interval |
| **DP** | Depersonalization |
| **EE** | Emotional Exhaustion |
| **GDPR** | General Data Protection Regulation |
| **HCW** | Healthcare workers |
| **MBI** | Maslach Burnout Inventory |
| **PPE** | Personal protective equipment's |
| **STROBE** | Strengthening the Reporting of Observational studies in Epidemiology |
| **VIF** | Variance Inflation Factor |
| **WMA** | World Medical Association |

## ACKNOWLEDGEMENTS

The authors thank the healthcare professionals for their responses and invaluable time. The authors acknowledge the management and reception office members of the healthcare centers at Jaipur division for the provision of the mail ID and mobile number of healthcare workers.

### Funding

This study is supported *via* funding from Prince Sattam bin Abdulaziz University project number (PSAU/2024/R/1445). The funders had no role in study design, data collection and analysis, decision to publish, or preparation of the manuscript.

### Grant Disclosures

The following grant information was disclosed by the authors:
Prince Sattam bin Abdulaziz University: PSAU/2024/R/1445.

### Competing Interests

Faizan Zaffar Kashoo is an Academic Editor for PeerJ.

### Author Contributions

- Mohammad Sidiq conceived and designed the experiments, performed the experiments, authored or reviewed drafts of the article, and approved the final draft.
- Sai Jaya Prakash Ch conceived and designed the experiments, authored or reviewed drafts of the article, and approved the final draft.
- Balamurugan Janakiraman conceived and designed the experiments, performed the experiments, analyzed the data, prepared figures and/or tables, authored or reviewed drafts of the article, and approved the final draft.

- Aksh Chahal conceived and designed the experiments, performed the experiments, analyzed the data, authored or reviewed drafts of the article, and approved the final draft.
- Imran Khan conceived and designed the experiments, performed the experiments, authored or reviewed drafts of the article, and approved the final draft.
- Surbhi Kaura conceived and designed the experiments, performed the experiments, analyzed the data, prepared figures and/or tables, authored or reviewed drafts of the article, and approved the final draft.
- Faizan Kashoo conceived and designed the experiments, analyzed the data, prepared figures and/or tables, authored or reviewed drafts of the article, and approved the final draft.
- Farha Khan conceived and designed the experiments, authored or reviewed drafts of the article, and approved the final draft.
- Shabnam Khan conceived and designed the experiments, performed the experiments, prepared figures and/or tables, authored or reviewed drafts of the article, and approved the final draft.
- Chhavi Arora Sehgal conceived and designed the experiments, performed the experiments, prepared figures and/or tables, authored or reviewed drafts of the article, and approved the final draft.
- Shashank Baranwal conceived and designed the experiments, performed the experiments, authored or reviewed drafts of the article, and approved the final draft.
- Sheenam Popli conceived and designed the experiments, performed the experiments, authored or reviewed drafts of the article, and approved the final draft.
- Mshari Alghadier conceived and designed the experiments, analyzed the data, authored or reviewed drafts of the article, and approved the final draft.

## Human Ethics

The following information was supplied relating to ethical approvals (*i.e.*, approving body and any reference numbers):

Departmental Ethics Committee Nims College of Physiotherapy & Occupational Therapy, NU/NCPT/JUNE/17

## Ethics

The following information was supplied relating to ethical approvals (*i.e.*, approving body and any reference numbers):

Departmental Ethics Committee Nims College of Physiotherapy & Occupational Therapy, NU/NCPT/JUNE/17

## Data Availability

Data has been uploaded to repository (figshare).

https://figshare.com/articles/dataset/Data_set_and_Code_book_of_COVID/25772415/1.

## Supplemental Information

Supplemental information for this article can be found online at http://dx.doi.org/10.7717/peerj.18039#supplemental-information.

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
