# Peer review of "Burnout experience among healthcare workers post third COVID-19 wave in India; findings of a cross-sectional study"

_PeerJ, doi:10.7717/peerj.18039_

## Round 0.1 · original submission · Major Revisions

Please address to the comments from the esteemed reviewers and also do not miss the attached file by one of the reviewers.

Reviewer 1 ·

Basic reporting

Introduction and Background:

The introduction provides a good overview of the context and significance of the study. However, it could be improved by providing more specific details about the knowledge gap being addressed and how the study aims to fill it.
For example, you could provide more information about the current state of burnout among healthcare workers in India, the impact of the COVID-19 pandemic on their mental health, and how the study's findings will contribute to the existing body of knowledge.

Structure and Organization:
The structure of the paper is generally clear and easy to follow. However, some sections could be improved by providing more specific details and examples to support the discussion.
For instance, the background section could benefit from more recent references to support the discussion on burnout among healthcare workers. This would help to update the understanding of burnout, enhance the context, and support the study's findings.

Language and Grammar:
The language is generally clear and professional, but there are some areas where the phrasing could be improved for better clarity and comprehension.

Ambiguous Phrasing: For example, in the Background section, the sentence "The mental health of healthcare workers was a public health issue even before COVID-19 and the pandemic increased healthcare workers' stress and burnout" could be rephrased to clarify the relationship between the pandemic and the mental health issue. Instead of using "even before COVID-19," the authors could specify the time period before the pandemic when the mental health issue was already a concern.

Complex Sentences: In the Methods section, the sentence "A cross-sectional study was conducted among different healthcare workers from multi-healthcare centers engaged in COVID-19 management in Jaipur division, Rajasthan State" could be broken down into simpler sentences for better clarity. This would help readers understand the study design and population more easily.

Technical Terms: In the Results section, the sentence "High burnout scores were linked to nursing profession (³ = 7.89, 95% CI; 3.66, 12.11, p < 0.0001)" could be rephrased to provide more context about the statistical analysis. For instance, the authors could explain what the p-value and confidence interval represent in simpler terms.

Sentence Structure: In the Discussion section, the sentence "Health workers are still burnt out due to personal, professional, and patient factors, requiring organizational and individual interventions" could be rephrased to improve sentence structure. Instead of using a long, complex sentence, the authors could break it down into simpler sentences or use shorter sentences with clear subject-verb-object structures.

Experimental design

Study Design:
The study design is well-justified and appropriate for the research question. However, it could be improved by providing more specific details about the sampling strategy and how the participants were selected.
For instance, you could explain how you obtained the list of email IDs of potential healthcare professionals, how you ensured that the participants were representative of the target population, and how you handled non-response rates.

Methods:
The methods section is generally well-written and provides sufficient detail for replication. However, some sections could be improved by providing more specific details about the data collection and analysis procedures.
For example, you could provide more information about the specific recruitment process, such as how healthcare professionals were initially contacted and invited to participate, what information was included in the recruitment materials, and how you ensured that the sample was representative of the target population.

Data Analysis:
The data analysis is well-described and appropriate for the research question. However, it could be improved by providing more specific details about the statistical tests used and how the results were interpreted.
For instance, you could explain the type of regression model used, the assumptions checked to ensure the appropriateness of the regression analysis, and how you handled any missing data or outliers in the dataset.

Validity of the findings

Impact and Novelty:
The study provides new insights into the levels of stress and burnout among healthcare workers in India following the third wave of COVID-19. However, the impact and novelty of the findings could be improved by providing more specific details about the implications for healthcare policy and practice.
For example, you could suggest strategies for improving work-life balance, enhancing job security, and increasing support for healthcare workers.

Replication:
The study provides a comprehensive analysis of the data, but it could be improved by providing more specific details about the replication of the results and how they relate to other studies in the field.
For instance, you could explain how you checked for consistency across different subsamples, how you handled missing data or outliers, and how you ensured the robustness of the findings.

Conclusion:
The conclusion is generally well-written and summarizes the main findings of the study. However, it could be improved by providing more specific details about the implications for healthcare policy and practice and how the findings relate to other studies in the field.
For example, you could discuss how your results compare to those of other studies that have examined burnout among healthcare workers during the COVID-19 pandemic, and how your findings contribute to the existing body of knowledge on this topic.

Additional comments

General Comments

Strengths:
The study provides a comprehensive analysis of the levels of stress and burnout among healthcare workers in India following the third wave of COVID-19. The study also provides new insights into the predictors of burnout and the implications for healthcare policy and practice.

Weaknesses:
The study could be improved by providing more specific details about the sampling strategy and how the participants were selected. Additionally, the study could benefit from more recent references to support the discussion on burnout among healthcare workers.
Suggestions for Improvement:
The study could be improved by providing more specific details about the data collection and analysis procedures. Additionally, the study could benefit from more specific details about the implications for healthcare policy and practice and how the findings relate to other studies in the field.

·

Basic reporting

The title is: Stress and burnout experience among healthcare workers post third COVID-19 wave in India; findings of a cross-sectional study, but there is no data about STRESS experience in this manuscript. As we know stress experience can measure using perceived stress scale (PSS).

I think the findings of this study can not be generalized to all health care professional in India because the data were collected from one of the healthcare centers in the Jaipur division

Experimental design

In Table 2 we could find analysis of burn-out and socio-demographic characteristics with variables of: number of children and BMI category, but
In Table 1 we could not find the data (number of children and BMI category).., Why??

Validity of the findings

(P. 284). The authors stated that: Burnout was found to be associated with some of the demographic factors. Significant gender disparities were observed, as males reported higher levels of personal burnout in comparison to female healthcare workers.

In discussion the authors did not analyze the statement deeply, the authors should explain why male have higher burnout levels than female.

The authors should also analyze another demographic factor: age.

Additional comments

-

---

## Round 0.2 · accepted · Accept

Congratulations and thank you for addressing to the comments raised

Reviewer 1 ·

Basic reporting

no comments

Experimental design

no comments

Validity of the findings

no comments

Additional comments

i am satisfied with the revisions done by the authors